# How Social Norms Affect Consumer Intention to Purchase Certified Functional Foods: The Mediating Role of Perceived Effectiveness and Attitude

**DOI:** 10.3390/foods10061151

**Published:** 2021-05-21

**Authors:** Edward Shih-Tse Wang, Yun-Hsuan Chu

**Affiliations:** Graduate Institute of Bio-Industry Management, National Chung Hsing University, 250, Kuo Kuang Rd., Taichung 402, Taiwan; Yun-Hsuan.Chu.NCHU@gmail.com

**Keywords:** certified functional foods (CFFs), descriptive norms, injunctive norms, attitude, perceived effectiveness

## Abstract

Certified functional foods (CFFs) are approved by relevant authorities because of demonstrable efficacy. However, social norms affect consumer perceptions regarding CFFs, and their attitudes toward CFFs remain unclear. Drawing on social influence theory, this study explored how social norms (i.e., descriptive and injunctive) affect consumer perceptions and willingness to purchase CFFs. Consumers of CFFs in Taiwan were invited to participate in this study, and 398 valid questionnaires were received. Collected data were assessed through structural equation modeling. The results revealed that descriptive and injunctive norms exerted a positive effect on perceptions of the effectiveness of CFFs. However, although injunctive norms exerted a positive effect on consumer attitude, the effect of descriptive norms on attitude was not significant. Furthermore, consumer perceptions on the effectiveness of CFFs affected their attitude toward CFFs, consequently increasing their intention to purchase CFFs. This study contributes to the literature by providing insights into the relationship between social norms, perceived effectiveness, and attitudes regarding CFFs. The results of this study provide directions to CFF marketers for developing marketing strategies and establishing marketing communication strategies from the perspective of social influence.

## 1. Introduction

Because of growing concern over the consequences of poor dietary habits, including cancer, cardiovascular diseases, overweight and obesity, and diabetes, consumers interest in functional foods is increasing [1]. Chen [2] reported that several factors increase the willingness of consumers to buy functional foods, including their awareness of diet-related health problems, their degree of busyness, the convenience of food choices, and whether they exercise. Functional foods, exemplified by dietary supplements, fortified foods, designer foods, and healthy foods, are typically enriched with nutrients and vitamins that are beneficial for health [3]. Thus, functional foods fulfill basic nutritional requirements [4], enabling the consumption of a balanced and healthy diet and preventing many diseases [5]. Consumer awareness regarding the health effects of functional foods has stimulated the growth of the functional food market [6]. However, because some functional food consumers may be skeptical of the capacity of efficient meals to deliver on the promise of health benefits, the functional food manufacturing industry has focused on the development of certified functional foods (CFFs) [7]. Given that an increasing number of people are seeking safe and healthy food alternatives [8], the market for CFFs is growing worldwide [9].

CFF marketers display health claims on food labeling to enhance consumer perception of the effectiveness of the product and engender a positive attitude, with the hope that positive perceptions and attitudes translate into sales. However, individuals sometimes make purchasing decisions after rational thinking, and such decision making is influenced by norms [10]. Osterhus [11] reported that behavior is affected by norms. Norms refer to an individual’s own expectations as well as the expectations of society [12]. Perceived social expectations influence the behavior of individuals [13]. Social norms refer to the expectations of appropriate behavior in a group environment; these are internalized by individuals and guide their behavior [14]. One study indicated that social norms influence consumer attitudes and purchasing intentions with respect to sustainable foods [15]. In addition, another study reported that social norms are highly influential in consumer decision making in general and can be wielded extremely effectively in promoting healthy food choice behavior in particular [16]. Social norms can be divided into two types: descriptive and injunctive [17]. Descriptive norms refer to an individual’s perception of the particular behavior of others, whereas injunctive norms refer to an individual’s perception of other people’s approval or disapproval of a certain behavior [18]. Many food consumption researchers have examined the effects of descriptive and injunctive norms on consumer decisions regarding the consumption of fruits and vegetables [19,20], snacks [21], eco-friendly food [22], and local foods [23].

Nystrand and Olsen [24] proposed a conceptual framework suggesting that factors from the theory of planned behavior (i.e., social norms (injunctive norms), attitude, and perceived behavioral control), self-efficacy, and descriptive norms affect consumer intention to eat functional foods and determined that descriptive and injunctive norms as well as attitude directly and significantly affected consumer intention to eat functional foods. In addition to the direct relationship between norms and attitude, another study suggested that attitude mediates the direct effect of social norms on food consumption intentions [25]. Furthermore, the perceived effectiveness of functional food refers to the extent to which consumers believe that consuming functional foods benefits their health. The perceived effectiveness of intervention strategies for encouraging healthy food choices is related to consumer acceptance of such strategies [26]. However, whether injunctive and descriptive norms regarding functional foods are related to the perceived effectiveness of functional foods and consumer attitude toward functional food consumption remains unclear. Because social norms can effectively promote healthy choices and behaviors and are crucial in consumer decision making [16] and the extent to which the value of functional foods is recognized is increasing, developing a better understanding of functional food consumption behavior from the perspective of social influence is crucial for marketers.

## 2. Literature Review and Research Hypotheses

Higgs [27] contended that diet often depends on the social environment and that the influence of people important to them may affect individuals’ perceptions and evaluations. Social scientists have emphasized the importance of social influence in human behavior, and social norms are a crucial factor in social influence. Social norms affect people’s buying behavior [28]. In the domain of social psychology, researchers argue that descriptive and injunctive norms affect behavior in groups [29]. However, descriptive and injunctive norms differ [30]. Descriptive norms refer to the behavioral expectations of most people in society, whereas injunctive norms refer to the approval or disapproval of behaviors by people who influence an individual [31]. Studies have reported that descriptive and injunctive norms affect the motivation to protect a hotel’s environment [32], adolescents’ attraction to and use of marijuana [33], pregnant women’s dietary choices [34], and consumer intention to purchase functional foods [24]. Moreover, studies have indicated that the effects of injunctive and descriptive norms may be aligned; however, their effects may conflict in some instances [18,35].

### 2.1. Effect of Descriptive Norms on Consumer Perceptions of the Effectiveness of CFFs and Consumer Attitude toward CFFs

Descriptive norms refer to the behavior of other people in a social environment. Perceived effectiveness reflects the understanding of the potential positive consequences of engaging in a particular behavior [36]. Descriptive norms provide information to people regarding how to appropriately act in certain situations to achieve particular goals [37]. Moreover, a study suggested that descriptive norms affect the perceived effectiveness of environmentally friendly behavior [38]. Where individuals feel that adhering to social nutritional norms can improve their health, they tend to look to the consumption behaviors of others as a guide to their own food decisions and consume what others consume as a means of fulfilling the need to consume safe and healthy food [27]. On the basis of the preceding discussion, the following hypothesis was proposed:

**Hypothesis** **1** **(H1).***Descriptive norms exert a positive effect on the perceived effectiveness of CFFs*.

Experimental evidence suggests that descriptive norms affect people’s food choices [39]. Others making choices can be regarded as their possible expression of social rules in a given context; thus, descriptive norms can affect certain behaviors [40]. Attitude refers to the set of beliefs regarding a behavior that is based on an evaluation of the results of the behavior [41]. According to Kim and Chung [42], consumers are interested in buying a product if they think that influential people believe the product to be good. On the basis of these findings, the following hypothesis was proposed:

**Hypothesis** **2** **(H2).***Descriptive norms exert a positive effect on consumer attitude toward CFFs*.

### 2.2. Effect of Injunctive Norms on Perceived Effectiveness of CFFs and Consumer Attitude toward CFFs

Injunctive norms refer to an individual’s belief regarding whether other people believe that the individual is acting in accordance with certain standards or complying with social pressure [12]. Perceived effectiveness is related to the concept of cognitive behavioral control [43]. A study reported that beliefs concerning injunctive norms are related to self-efficacy [44]. An individual’s perceived effectiveness of functional foods reflects their belief that functional foods can positively affect or change their health. A study reported that injunctive norms positively affect the perceived effectiveness of environmentally friendly behavior [38]. Therefore, if people who are important to a consumer approve of the consumption of functional foods, then the consumer will consider such functional foods to be effective. Accordingly, the following hypothesis was proposed:

**Hypothesis** **3** **(H3).***Injunctive norms exert a positive effect on the perceived effectiveness of CFFs*.

Injunctive norms refer to the approval or disapproval of a certain behavior by other important people [45]. Individuals form their attitude toward a product on the basis of collective pressure exerted by others. Injunctive norms can predict the consumption of functional foods and the pursuit of a healthy diet [30]. Han [46] found that injunctive norms strongly influenced attitude formation. Accordingly, the following hypothesis was proposed:

**Hypothesis** **4** **(H4).***Injunctive norms exert a positive effect on consumer attitude toward CFFs*.

### 2.3. Effect of the Perceived Effectiveness of CFFs on Attitude and Purchase Intention

Consumers buy certain products to satisfy certain desires [47]. Consumers compare and select products that best meet their needs or solve their problems [48]. Vermeir and Verbeke [15] indicated that perceived effectiveness is necessary to motivate consumers to have a positive attitude toward a product, and perceived effectiveness can exert a significant and positive effect on the attitude of consumers toward the purchased product. In addition, studies have confirmed that consumer beliefs regarding perceived effectiveness are positively related to their attitude toward products [47]. Perceptions of high effectiveness can result in positive consumer attitudes [15,43]. On the basis of these findings, the following hypothesis was proposed:

**Hypothesis** **5** **(H5).***Perceived effectiveness exerts a positive effect on consumer attitude toward CFFs*.

### 2.4. Effect of Attitude on Purchase Intention

Many studies have explored the relationship between consumer attitude and purchase intention [42,49]. According to Ajzen’s [28] theory of planned behavior, people who have a favorable attitude toward a particular behavior are prone to exhibiting such a behavior. Hence, the more positive a consumer’s attitude toward a functional food is, the stronger their intention to purchase the functional food will be. Accordingly, the following hypothesis was proposed:

**Hypothesis** **6** **(H6).***Attitude toward a functional food positively affects their intention to purchase that CFF*.

On the basis of the aforementioned seven hypotheses, the framework of this study is presented in Figure 1.

## 3. Materials and Methods

### 3.1. Ethics Statement

The research protocol of this study was not submitted for approval from a certified institutional review board (IRB) because, in Taiwan, IRB approval is required only for natural science research involving animal or human participants and not for social science research. Nonetheless, the research protocol was designed in accordance with the Ethical Guidelines for Research with Human Subjects of the American Psychological Association. In this study, individuals in public places were approached and asked if they would voluntarily participate in this study’s survey; no financial incentive was provided. The participants were briefed regarding the research aims before enrolling, and they were assured that their responses would be anonymized (Appendix A). We also considered a completed questionnaire as provision of written informed consent. With regard to anonymity, the responses did not contain individual identifiers and could not be traced back to any individual.

### 3.2. Measurement

This study investigated the effects of social norms on consumer perceptions of effectiveness, attitude, and behavioral intention. A questionnaire was used that was split into two components. The first component consisted of questions related to dimension items for five variables. Descriptive norms were measured using a modified version of the 3-item scale developed by Smith et al. [35] (sample item: You think people who are important to you are very likely to eat CFFs), and injunctive norms were measured using a 4-item scale modified from the scale developed by Smith et al. [35] (sample item: Your consumption of CFFs is in line with what most people expect). Perceived effectiveness was examined using a 4-item scale developed by Wang and Lin [38] with some modifications (sample item: You believe that eating CFFs will benefit your body). Attitude was evaluated using a modified version of a 4-item scale developed by Song et al. [50] (sample item: In general, you are positively disposed toward CFFs). Purchase intention was measured using a 4-item scale developed by Bian and Forsy [49] with some modifications (sample item: The likelihood that you would purchase CFFs is high). Participants’ responses for all items included in these scales were scored on a 7-point Likert scale, with a score of 1 indicating strong disagreement and a score of 7 indicating strong agreement. The second component of the questionnaire included questions related to the personal data of respondents, including gender, age, educational level, and monthly disposable income.

### 3.3. Data Collection and Sampling

The research was conducted in Taiwan. The homogenous sociodemographic characteristics of participants (i.e., consumers in Taiwan) allowed us to control for sociocultural effects. Taiwan’s government has mandated that for food manufacturers to market their functional foods as having health benefits, the food must bear a certification label (i.e., health label) from Taiwan’s Department of Health [7]. Questionnaires were distributed at Taichung railway station. Because no statistics were available to develop a sampling framework for the population consuming CFFs (e.g., age and sex) and probability sampling was unavailable, convenience sampling was used to collect data at a railway station. A railway station was selected for questionnaire distribution because the population size and distribution of CFF consumers in Taiwan are unknown and because railway stations are public places frequented by individuals representing a broad spectrum of the population [51]. Consumers who had consumed functional foods with a certification mark were included as participants in this study. A total of 684 people were invited to participate in the survey; from these 684 people, 400 valid questionnaires were obtained after responses to the following filter question were discarded: “Have you ever eaten functional food with the certification mark?” After the exclusion of two incomplete questionnaires, 398 valid samples were collected. Because the sample size should be 20 times larger than the number of observation items in multivariate research [52] and 19 items were used in the study, the necessary sample size was larger than 380; 398 valid samples satisfied this requisite minimum sample size. Of the 398 samples, 257 (64.6%) respondents were women and 141 (35.4%) were men. Because women tend to have, in general, a strong propensity to purchasing healthy food [53], the sex ratio was considered acceptable in this survey. The majority of respondents (32.7%) were aged 20–29 years, and 101 (25.4%) respondents each were aged 30–39 and 40–49 years. Furthermore, 233 (58.5%) and 136 (34.2%) respondents had a university degree and a master’s degree or above, respectively. A total of 109 (27.4%), 51 (12.8%), and 44 (11.1%) respondents had a monthly disposable income of TWD 40,001, TWD 5001–10,000, and TWD 35,001–40,000, respectively. Regarding health food types, 208 (52.3%), 172 (43.2%), and 18 (4.5%) respondents consumed health foods through capsules, beverages, and other forms, respectively.

## 4. Data Analysis

The statistical software LISREL version 8.80 and structural equation modeling (SEM) were utilized to examine the relationship between observed and potential variables and analyze the causality between the variables. SEM can be used to assess measurement and structural models for performing factor and path analyses. The measurement model can be used to examine the relationship between potential and observed variables, and confirmatory factor analysis (CFA) can be conducted to measure the reliability, validity, and fit of the model. A 19-item CFA was performed to examine whether measured variables reliably reflected hypothetical potential variables. Convergent validity was evaluated according to the following criteria reported by Fornell and Larcker [54]: factor loading estimates should be >0.5, composite reliability (CR) should be >0.6, and the average variance extracted (AVE) should be >0.5. Moreover, a measurement model is also considered to exhibit discriminant validity if the degree of the relationship between potential variables is lower than the degree of the relationship within observed variables. Discriminant validity can be evaluated by examining the relationship matrix between constructs. The square root of the AVE of latent variables should be greater than the correlation coefficient of other distinct constructs.

## 5. Results

As shown in Table 1, the factor loading (λ), AVE values, and CR values of each item ranged from 0.83 to 0.95, from 0.73 to 0.85, and from 0.91 to 0.97, respectively, indicating that the aforementioned criteria for internal reliability and convergent validity were satisfied.

As shown in Table 2, the square root of the AVE of each variable was greater than the correlation coefficient between constitutive variables, indicating the presence of discriminant validity.

The results for the overall fit indices in this study were as follows: absolute fit measure: χ^2^ = 506.78, df = 145, χ^2^/df = 3.50, goodness-of-fit index (GFI) = 0.89, adjusted GFI = 0.85, and root mean square error of approximation = 0.077; comparative fit measure: non-normed fit index = 0.98 and comparative fit index = 0.99; and parsimonious fit measure: parsimonious normed fit index = 0.83 and parsimonious GFI = 0.68. The fit of the overall structural model met the standard values recommended by scholars [55]. Therefore, the overall fit of this study model was acceptable.

The path analysis results revealed that descriptive and injunctive norms exert a significantly positive effect on perceived effectiveness, thus supporting H1 and H3. Injunctive norms exerted a positive effect on attitude, whereas descriptive norms exerted no significant effect on attitude; thus, these results supported H4 but not H2. Perceived effectiveness exerted a positive effect on attitude, thus supporting H5. Attitude exerted a positive effect on purchase intention, thus supporting H6. Furthermore, the results indicated that the model explained 44%, 75%, and 74% of perceived effectiveness, attitude, and purchase intention, respectively. The hypothesis test results are presented in Table 3.

To verify the indirect model, an alternative model was used in which the direct effects of descriptive norms, injunctive norms, perceived effectiveness, and attitude on the intention to purchase functional foods were investigated. This model was based on the findings of Nystrand and Olsen [24], who reported that descriptive and injunctive norms as well as attitude directly and significantly affected consumers’ intention to eat functional foods. The alternative model was a direct one. To evaluate the two models, the Akaike information criterion (AIC) along with the consistent AIC (CAIC) were used. Lower values of the AIC and CAIC indicated better fit of the model to the data [56]. In the indirect model, the AIC was 596.78, and the CAIC was 823.37. In the alternative model (direct model), the AIC was 598.22, and the CAIC was 839.93. These results indicated that the indirect model was superior to the direct model. In addition, the results for the alternative model revealed that descriptive norms, injunctive norms, and perceived effectiveness did not significantly affect consumer purchase intention.

Baron and Kenny’s (1986) [57] four-step test was used to test our hypothesis that the perceived effectiveness of CFFs mediates the effect of descriptive and injunctive norms on attitude toward CFFs and that attitude toward CFFs mediates the effect of descriptive norms, injunctive norms, and perceived effectiveness of CFFs on purchase intention. As shown in Table 4, the perceived effectiveness of CFFs fully mediated the effects of descriptive norms on attitude toward CFFs, whereas the perceived effectiveness of CFFs partially mediated the effects of injunctive norms. Furthermore, attitude toward CFFs fully mediated the effects of descriptive norms, injunctive norms, and perceived effectiveness of CFFs on purchase intention.

## 6. Discussion

This study investigated the antecedents of consumer CFF purchase intention from the perspective of social norms. A conceptual model was developed to examine the effect of descriptive and injunctive norms on consumers’ perceptions of effectiveness, attitude, and purchase intention. The results revealed that consumers’ attitudes toward CFFs exerted a positive effect on their CFF purchase intention; this finding is consistent with those of previous studies, which indicated that Danish and Norwegian consumers’ intentions to purchase functional foods are affected by their attitude toward functional foods [24,58]. Next, perceived effectiveness exerted a positive effect on attitude toward CFFs. This result is consistent with that reported by Landström et al. [59], who documented that the perceived effect of functional foods is related to attitudes toward functional foods. In addition, no study has attempted to explore the role of social norms in the perceived effectiveness of functional foods. This research is one of the first to investigate the effect of social norms on the perceived effectiveness of functional foods, and the results of this study indicated that descriptive and injunctive norms affect the perceived effectiveness of CFFs. This finding is consistent with that of a previous study, which reported that prosocial norms (i.e., descriptive and injunctive norms) exert a positive effect on the perceived effectiveness of environmentally friendly behaviors [38]. Finally, little is known regarding how social norms influence consumers’ attitude toward CFFs. Thus, the present study is the first study to examine the relationship between social norms and consumers’ attitude toward CFFs. Lac and Donaldson [60] found that injunctive norms from both friends and parents significantly influence college students’ attitudes toward alcohol; however, in terms of descriptive norms, their attitudes were only affected by their friends and not by their parents. This result revealed that although the effects of descriptive and injunctive norms are similar, they do differ due to formation through social interaction with different reference groups. Thus, our finding is consistent with their study. That is, injunctive norms, but not descriptive norms, directly affected consumers’ attitude toward CFFs. No study has attempted to elucidate possible mediating relationships among descriptive norms, injunctive norms, and consumer perceptions of effectiveness, attitude, and purchase intention. This study contributes to the literature through its investigation of the relationships among social norms, perceived effectiveness, and attitude. Our findings promote an understanding of the mediating roles of the perceived effectiveness of CFFs and attitude toward CFFs. The results indicate that descriptive and injunctive norms exert an indirect effect on consumers’ willingness to purchase CFFs through perceived effectiveness and attitude.

The results of this study demonstrated that consumer perceptions of the effectiveness of CFFs affected their attitude toward CFFs. Thus, to facilitate consumer consumption, marketers should seek to foster a positive attitude among consumers regarding the effectiveness of their foods. Where CFFs have demonstrated health benefits and have been approved by the responsible government agency, marketers should emphasize the health benefits of their products in marketing materials and highlight the approval by the government agency to enhance consumer perceptions of the effectiveness of their products. Additionally, our findings suggest that functional food marketers can enhance consumer perceptions of the effectiveness of CFFs by leveraging the power of social influence. Our results indicated that injunctive norms affect the perceived effectiveness of CFFs, consequently resulting in the formation of a positive attitude toward CFFs and increasing willingness to purchase CFFs. Thus, according to our findings, functional food marketers can use social influence strategies to influence consumer perceptions of the effectiveness of CFFs and engender a positive attitude toward CFFs. For instance, injunctive norms become operative when an individual receives product recommendations from friends or family, which can further increase the perceived effectiveness of CFFs, thus resulting in a positive attitude and purchase intention. To retain existing consumers, health food marketers can design marketing strategies to encourage existing consumers to recommend products to their friends or family. Moreover, increased visibility can promote the formation of descriptive norms, increase awareness, and provide more opportunities to convey the perceived effectiveness of CFFs and inspire a positive attitude toward such products. Thus, health food marketers can design small and portable packages for consumer use in public; this can increase other consumers’ awareness of CFFs.

This study has some limitations. First, because of the convenience sampling method and the relatively small sample size (398 CFF consumers) in this study, the generalizability of the results may be limited. Convenience sampling may not be representative of the targeted population, and questionnaires were distributed in only one public place located in central Taiwan (i.e., Taichung railway station). Therefore, further research should include larger, nationwide samples from various locations covering northern, southern, and eastern Taiwan to enhance sample representativeness and the generalizability of findings. Second, samples were collected within a specific region (Taiwan). Although this allowed us to control for sociocultural effects, the results may not be particularly representative of populations in other countries. Moreover, because society in Taiwan is, in general, compliant [61], the effects of social norms may have been amplified. Researchers have indicated that differences in complaining behavior are explained in terms of underlying cultural values (e.g., individualism vs. collectivism) [62]. As Western values (e.g., the United States) are mostly individualistic, whereas eastern values (e.g., Taiwan) are mostly collectivistic [63,64], future studies should replicate our findings in other countries and cultural settings to improve the value of the research. Third, this study used social norms as an independent variable. Follow-up studies can examine various other factors such as product attributes (e.g., country of origin), personal characteristics (e.g., skepticism), and problematic situations (e.g., pandemic period) to understand functional food consumption behaviors. It is important to understand various other factors that may increase consumers’ confidence in the effectiveness of CFFs. Fourth, as the functional foods market includes many non-CFFs, future research is required to replicate our findings in non-CFFs settings. Moreover, an investigation into the influence of social norms on perceived effectiveness, attitude, and purchase intention among CFF and non-CFF consumers could provide valuable insights. Additional studies are warranted to help functional food marketers develop more effective marketing programs.

## Figures and Tables

**Figure 1 foods-10-01151-f001:**
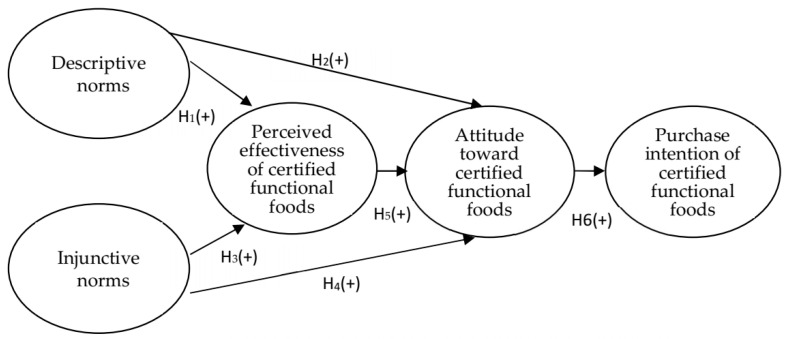
Research framework and results.

**Table 1 foods-10-01151-t001:** Accuracy analysis statistics.

Research Constructs	Measurement Items	Factor Loading	C.R.	AVE
Descriptive norms	1. You think that many people who are important to you eat certified functional food.	0.88	0.92	0.80
	2. You think people who are important to you are very likely to eat certified functional food.	0.91		
	3. You think that a large proportion of people who are very important to you also eat certified functional food.	0.90		
Injunctive norms	1. People who are very important to you recognize your consumption of certified functional food.	0.83	0.91	0.73
	2. Your consumption of certified functional foods is in line with what most people expect.	0.86		
	3. Most people who are very important to you endorse your consumption of certified functional food.	0.88		
	4. You and many people who are very important to you think that eating certified functional food is good.	0.84		
Perceived effectiveness	1. You believe that if you eat certified functional foods, it will have a positive impact on your body.	0.93	0.96	0.85
	2. You believe that eating certified functional food will benefit your body.	0.94		
	3. You believe that eating certified functional food will improve your physical health.	0.91		
	4. You believe that if you eat certified functional food, your bodily functions will be improved.	0.91		
Attitude	1. You think that certified functional food is worth consuming.	0.88	0.93	0.77
	2. You like certified functional food.	0.84		
	3. In general, you are positively disposed toward certified functional food.	0.90		
	4. In general, you think that certified functional food is good.	0.88		
Purchase intention	1. The probability you would consider buying certified functional food is high.	0.93	0.97	0.84
	2. You will consider buying certified functional food.	0.93		
	3. The likelihood that you would purchase certified functional food is high.	0.95		
	4. You are willing to buy certified functional food.	0.93		

Note: C.R.: Composite Reliability; AVE: Average Variance Extracted

**Table 2 foods-10-01151-t002:** Correlations between research constructs.

Research Constructs	Factor Loading	C.R.	AVE	Mean	S.D.	Correlation				
						DN	IN	PE	ATT	PI
Descriptive norms (DN)	0.88~0.91	0.92	0.80	6.11	1.13	**0.90**				
Injunctive norms (IN)	0.83~0.88	0.91	0.73	5.27	0.95	0.71	**0.85**			
Perceived effectiveness (PE)	0.91~0.94	0.96	0.85	5.41	0.96	0.53	0.66	**0.92**		
Attitude (ATT)	0.84~0.88	0.93	0.77	5.24	0.99	0.52	0.71	0.84	**0.88**	
Purchase intention (PI)	0.93~0.95	0.97	0.84	5.19	1.10	0.50	0.64	0.71	0.86	**0.92**

Note: C.R.: Composite Reliability; AVE: Average Variance Extracted; S.D.; Standard Deviation. The square root of the AVE for each construct is in bold (on the diagonal).

**Table 3 foods-10-01151-t003:** Hypothesis test results.

Hypothesis	Path between	Path Coefficients	*t*-Values
H1	Descriptive norms → Perceived effectiveness	+0.13 **	2.05
H2	Descriptive norms → Attitude	−0.40 ^n.s^	−0.96
H3	Injunctive norms → Perceived effectiveness	+0.56 ***	8.56
H4	Injunctive norms → Attitude	+0.32 ***	5.90
H5	Perceived effectiveness → Attitude	+0.66 ***	14.32
H6	Attitude → Purchase intention	+0.86 ***	21.67

*p <* 0.01 **, *p <* 0.001 ***, n.s = not significant.

**Table 4 foods-10-01151-t004:** Mediating effect test (Baron and Kenny’s (1986) approach).

Mediation Path			Path Coefficients					Mediating Effect
IV	M	DV	(1) IV → M	(2) M → DV	(3) IV → DV	(4) IV + M → DV		
						M → DV	IV → DV	
DN	PE	ATT	0.50 ***	0.79 ***	0.48 ***	0.63 ***	−0.02 ^n.s^	Full
IN	PE	ATT	0.62 ***		0.66 ***		0.28 ***	Partial
DN	ATT	PI	0.48 ***	0.82 ***	0.48 ***	0.69 ***	0.07 ^n.s^	Full
IN	ATT	PI	0.66 ***		0.60 ***		0.06 ^n.s^	Full
PE	ATT	PI	0.79 ***		0.69 ***		0.07 ^n.s^	Full

Note: IV = independent variable; M = mediator; DV = dependent variable; IN = injunctive norms; DN = descriptive norms; PE = perceived effectiveness; ATT = attitude; PI = purchase intention. n.s: not significant (*p* > 0.05); *** *p* < 0.001.

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
