# Peer review of "How Social Norms Affect Consumer Intention to Purchase Certified Functional Foods: The Mediating Role of Perceived Effectiveness and Attitude"

_foods, 2021, doi:10.3390/foods10061151_

Round 1
Reviewer 1 Report
*The last paragraph of the discussion section describes the limitation of this study and provides the strategy to overcome the limitation of this study as the suggestion of the “future studies”. However, the detailed information about the future studies (e.g. topic, contents, and/or study design) is omitted in the manuscript. Please explain more about the future studies based on the findings of this research.
*Discussion section should contain more cited references (i.e. previous researches in relevant areas) to describe the significance of this research. Implications of the major findings of this research should be derived by the analysis of more previous researches (e.g. major differences between the results of this study and previous relevant studies, and the reason for those differences such as the places for the survey, the time for the survey, and/or different sociodemographic characteristics)
*Please describe what (e.g. age, gender, etc.) and how (e.g. multistage stratified systematic sampling method, etc.) sociodemographic characteristics were considered to obtain the survey results from the participants with homogenous sociodemographic characteristics in the “Introduction” or “Materials and methods” section.
*Line 342: What is the “the generalizability the results”?
*Please carefully re-evaluate the manuscript to find the vague expression, grammar error, and typographical error.
Author Response
Q1-1. The last paragraph of the discussion section describes the limitation of this study and provides the strategy to overcome the limitation of this study as the suggestion of the “future studies”. However, the detailed information about the future studies (e.g. topic, contents, and/or study design) is omitted in the manuscript. Please explain more about the future studies based on the findings of this research.
A1-1. In response to the reviewer’s comment, suggestions regarding future studies have been added in detail in lines 349–368 of the revised manuscript.
If the reviewers find the data provided insufficient, more information can be added.
Q1-2. Discussion section should contain more cited references (i.e. previous researches in relevant areas) to describe the significance of this research. Implications of the major findings of this research should be derived by the analysis of more previous researches (e.g. major differences between the results of this study and previous relevant studies, and the reason for those differences such as the places for the survey, the time for the survey, and/or different sociodemographic characteristics)
A1-2. To address the reviewer’s concern, we have revised the manuscript. Please check lines 300–323 of the revised manuscript.
Q1-3. Please describe what (e.g. age, gender, etc.) and how (e.g. multistage stratified systematic sampling method, etc.) sociodemographic characteristics were considered to obtain the survey results from the participants with homogenous sociodemographic characteristics in the “Introduction” or “Materials and methods” section.
A1-3. The Data Collection and Sampling section has been revised. Please see lines 200–202 of the revised manuscript.
Q1-4.Line 342: What is the “the generalizability the results”?
A1-4. We have revised the sentence for clarity.
Q1-5.Please carefully re-evaluate the manuscript to find the vague expression, grammar error, and typographical error.
A1-5. The revised manuscript has been proofread by professional editors from Wallace Academic Editing. Furthermore, we have checked it thoroughly again.

Reviewer 2 Report
Thank you for the revision. I was unable to access the list of changes but find the revised paper acceptable.
Author Response
Q2-1. Thank you for the revision. I was unable to access the list of changes but find the revised paper acceptable.
A2-1. We are glad that the revised manuscript meets the reviewers’ expectations and standards.

Reviewer 3 Report
no further observation
Author Response
Q3-1. no further observation.
A3-1. We are glad that the revised manuscript meets the reviewers’ expectations and standards.
